# Common resting brain dynamics indicate a possible mechanism underlying zolpidem response in severe brain injury

**Shawniqua T Williams[1], Mary M Conte[1,2], Andrew M Goldfine[1], Quentin Noirhomme[3,4], Olivia Gosseries[3,4], Marie Thonnard[3,4], Bradley Beattie[5], Jennifer Hersh[6], Douglas I Katz[7], Jonathan D Victor[1,2], Steven Laureys[3,4], Nicholas D Schiff[1,2]\***

[1]Department of Neurology and Neuroscience, Weill Cornell Medical College, New York, United States; [2]Brain Mind Research Institute, Weill Cornell Medical College, New York, United States; [3]Coma Science Group, Cyclotron Research Centre, University of Liège and University Hospital of Liège, Liège, Belgium; [4]Neurology Department, University of Liège and University Hospital of Liège, Liège, Belgium; [5]Department of Neurology, Memorial Sloan Kettering Cancer Center, New York, United States; [6]Division of Medical Ethics, Weill Cornell Medical College, New York, United States; [7]Brain Injury Program, Braintree Rehabilitation Hospital, Boston University School of Medicine, Braintree, United States

**Abstract** Zolpidem produces paradoxical recovery of speech, cognitive and motor functions in select subjects with severe brain injury but underlying mechanisms remain unknown. In three diverse patients with known zolpidem responses we identify a distinctive pattern of EEG dynamics that suggests a mechanistic model. In the absence of zolpidem, all subjects show a strong low frequency oscillatory peak ~6–10 Hz in the EEG power spectrum most prominent over frontocentral regions and with high coherence (~0.7–0.8) within and between hemispheres. Zolpidem administration sharply reduces EEG power and coherence at these low frequencies. The ~6–10 Hz activity is proposed to arise from intrinsic membrane properties of pyramidal neurons that are passively entrained across the cortex by locally-generated spontaneous activity. Activation by zolpidem is proposed to arise from a combination of initial direct drug effects on cortical, striatal, and thalamic populations and further activation of underactive brain regions induced by restoration of cognitively-mediated behaviors.

**\*For correspondence:** nds2001@med.cornell.edu

**Competing interests:** The authors declare that no competing interests exist.

**Reviewing editor**: Eve Marder, Brandeis University, United States

## Introduction

Induced recovery of spoken language, cognitive and motor functions following administration of zolpidem, a gamma-aminobutyric acid type A (GABA-A) subclass alpha 1 receptor positive allosteric modulator (*Hambrecht-Wiedbusch et al., 2010*) in severely brain-injured subjects with disorders of consciousness are rare, but well documented (*Clauss et al., 2000*; *Brefel-Courbon et al., 2007*; *Cohen and Duong, 2008*; *Shames and Ring, 2008*; *Whyte, 2009*; *Hall et al., 2010*). In some instances, behavioral improvement spans the range from only limited signs of conscious awareness to recovery of conversant language, ambulation and coordinated motor activity within an hour of administration of the medication. However, the mechanisms underlying this phenomenon are poorly understood. Because of the rarity of the phenomenon, as well as logistical impediments, only a small number of prior studies have characterized zolpidem-induced physiological changes these patients (*Clauss et al., 2000*; *Clauss and Nel, 2006*; *Brefel-Courbon et al., 2007*; *Cohen and Duong, 2008*; *Shames and Ring, 2008*; *Whyte, 2009*; *Hall et al., 2010*; *Nyakale et al., 2010*).

**eLife digest** Some individuals who experience severe brain damage are left with disorders of consciousness. While they can appear to be awake, these individuals lack awareness of their surroundings and cannot respond to events going on around them. Few treatments are available, but a minority of patients show striking improvements in speech, alertness and movement in response to the sleeping pill zolpidem.

Although the idea of a sleeping pill increasing consciousness is paradoxical, it is possible that in patients with impaired consciousness, zolpidem reduces the activity of an area of the brain that would otherwise inhibit activity in other regions of the brain. However, the precise mechanisms by which zolpidem increases consciousness in these patients, and the reasons why only a minority of individuals respond, are unknown.

Now, Williams et al. have used electrodes attached to the scalp to measure changes in brain activity in three patients known to respond to zolpidem. These measurements showed that before the drug was taken, there were two important differences between the brain activity of the patients and that of healthy subjects: first, the patients showed brain waves of a lower frequency than any seen in healthy subjects; second, these brain waves were much more synchronized than brain activity in healthy individuals. After taking zolpidem, this synchronicity was reduced and all of the patients also showed an increase in higher frequency brain waves.

Based on the effects of zolpidem on electrical activity throughout the brain, Williams et al. propose a new model to explain the therapeutic action of the drug in some minimally conscious patients. If the correlation between brain waves and zolpidem response holds up in future studies, this relation could be used to predict which patients might benefit from the drug. A better understanding of these processes should also help us to understand, diagnose and develop new treatments for disorders of consciousness.

Here we investigate the detailed neurophysiological changes present in the resting electroencephalogram (EEG) in three patient subjects with such paradoxical behavioral facilitation in response to zolpidem and widely disparate underlying etiologies of severe brain injury and disorders of consciousness (severe hypoxic-ischemic encephalopathy, severe diffuse axonal injury and multifocal ischemic injuries). All subjects were more than 12 months post-injury and had histories of zolpidem response and regular usage of the drug for behavioral facilitation. Despite the wide differences in the etiology of brain injury across subjects, quantitative analysis of resting EEG dynamics reveals a striking commonality of spectral features in the absence of zolpidem, and stereotyped changes in these spectral quantities when the drug is administered.

Based on the specific spectral changes observed, we propose a unifying biological mechanism accounting for the observed dynamical features of the resting brain state and the changes in brain dynamics linked with zolpidem-induced behavioral facilitation.

Further, our interpretative model implies a mechanistic basis for the differences in EEG dynamics associated with the zolpidem response in subjects with global brain injury and those observed in a stroke patient (*Hall et al., 2010*), and offers predictions for distinguishing these two mechanisms.

## Results

### Behavioral effects of zolpidem in patient subjects

The three patient subjects were drawn from a wider observational study of recovery after severe brain injuries based on their history of behavioral facilitation with zolpidem; all subjects received zolpidem on a regular basis prior to this study (see 'Methods' Clinical Histories). As a first step, the zolpidem-induced behavioral responses were confirmed by formal assessments of behavior ON and OFF drug by the research teams ('Methods'). For subjects 1 and 3, this assessment was made with the Coma Recovery Scale-Revised, a well-established, standardized rating scale for patients with disorders of consciousness (*Giacino et al., 2004*; *Seel et al., 2010*); the CRS-R captures variations in cognitively-mediated behaviors up to a level of reliable and consistent communication through speech or gesture. For subject 2, baseline behavior was at ceiling for the CRS-R, so behavioral assessment was carried out by structured clinical observations.

All three subjects showed reproducible behavioral improvements following zolpidem. *Figure 1* shows CRS-R ratings (total and subscale) for Subject 1, following five doses (red lines) of the medication (four doses with concomitant video/EEG recordings studied here are labeled on Figure with red arrows). For assessments in the OFF-drug period (the first assessment, which followed a 62-hr washout, and at every time point that was at least 4 hr after zolpidem administration), the total CRS-R score ranged from 10–15. After each administration of zolpidem, the CRS-R score rose to ceiling (total score = 23). This change in CRS-R total score reflects improvements in all subscales, including recovery of functional movements, consistent communication, and elements of executive function. Additional changes not captured by this psychometric instrument included recovery of fluent verbal communication, writing and complex organized movements such as assembling block structures to match arbitrary configurations (see 'Clinical histories' for further clinical details and additional neuropsychological assessments). As seen in *Figure 1*, maximal total CRS-R scores consistently appeared within 1 hr after drug administration. The period during which the maximal CRS-R score was maintained appeared to

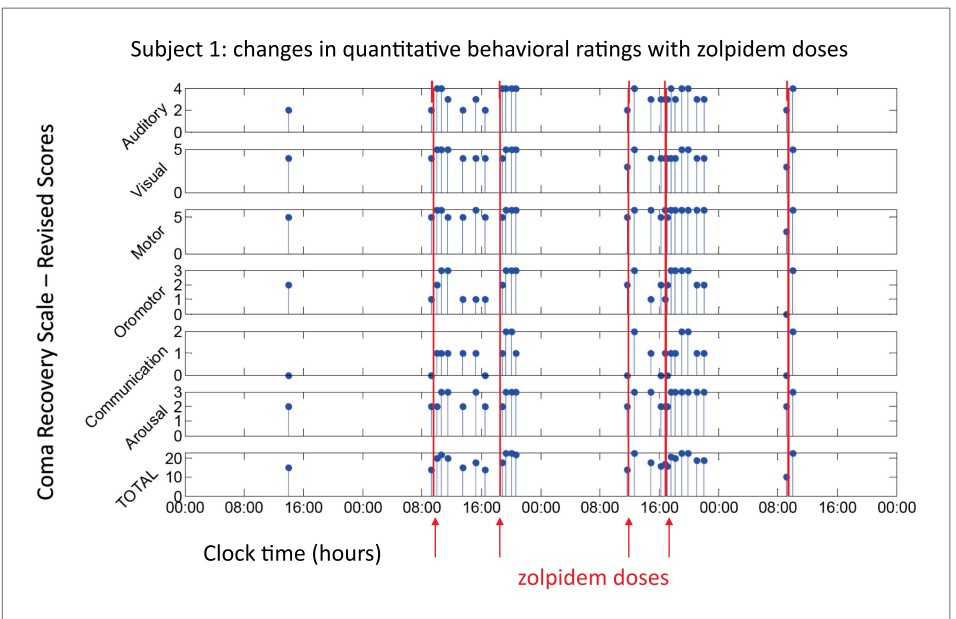

**Figure 1**. Behavioral changes associated with zolpidem doses in Subject 1. Subject 1 demonstrated only a limited range of behaviors in the baseline (OFF drug) state including automatic motor responses (e.g., reaching to a hand extended for a handshake, CRS-R motor subscore 5), oro-motor behaviors (e.g., opening mouth when presented with a spoon CRS-R oro-motor subscore 2 or biting a tongue depressor, CRS-R oro-motor subscore 1), localization of sound with head turning (CRS-R auditory subscale score 2), and reaching to objects (CRS-R visual subscale score 4). During all baseline assessments the patient demonstrated no evidence of command following or a communication system (CRS-R communication subscale score 0). Across three assessments of baseline behavior OFF drug after overnight periods and a 62 hr washout period at the onset of the study total CRS-R scores ranged from 10–15 reflecting a lack of goal-directed behaviors, evidence of any communication systems either verbal or gestural, nor consistent response to command following. Across two assessments of baseline behavior at least 4 hr after a prior dose of zolpidem within a day (reflecting typical duration of action of the medication) maximal total CRS-R scores of 18 reflected evidence of command following with inaccurate communication and higher level motor function (CRS-R subscale motor score of 6) or consistent auditory command following (CRS-R auditory subscale score of 4). Compared with these baseline behavioral assessments, consistent achievement of a maximum possible total CRS-R scores of 23 was obtained during all ON drug periods reflecting a state in which the patient consistently demonstrated behavioral levels not captured by this psychometric instrument including recovery of consistent communication, fluent verbal communication, writing and complex organized movements (see 'Clinical histories' for further clinical details and additional neuropsychological assessments). As seen in graph, a maximal total CRS-R scores consistently appeared following drug administration within approximately 1 hr and had a variable duration of maintenance with second daily dose of the medication showing extended time periods of maximal total scores. Red arrows indicate zolpidem doses for which accompanying EEG data were available for analyses. *Videos 1 and 2* illustrate aspects of the examinations to show correspondence of numerical ratings and behavior.

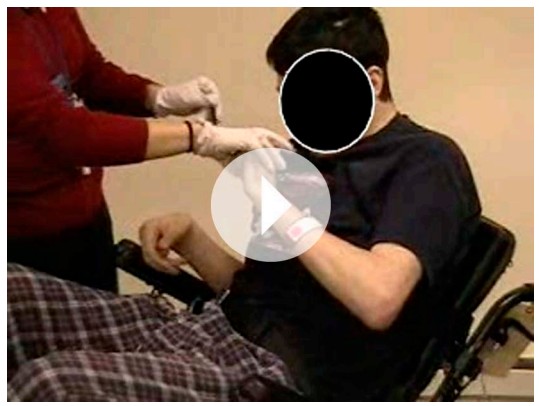

**Video 1**. Demonstration of CRS-R Motor subscale Subject 1 during off zolpidem state. Prior to zolpidem administration subject is tested with two common objects (comb and spoon) and asked to demonstrate their use. While able to hold each object he is unable to demonstrate their use. This level of behavior corresponds to a 5 on the CRS-R Motor subscale. Note resting tremor of right hand.

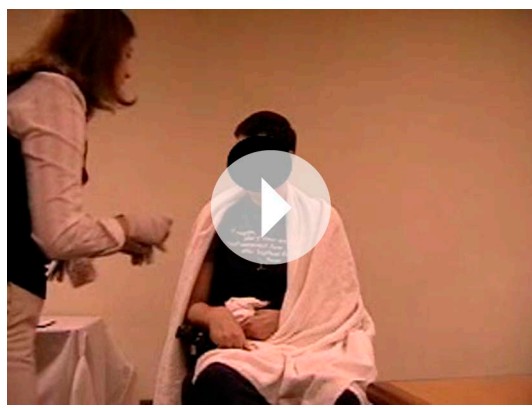

**Video 2**. Demonstration of CRS-R Motor subscale Subject 1 following zolpidem administration. Short legend: Following zolpidem administration subject is again tested with two common objects (comb and spoon) and asked to demonstrate their use. He is now able to demonstrate the use of each object. This level of behavior corresponds to a 6 on the CRS-R Motor subscale. Note that subject's posture has changed and that the right hand is used.

be longer for the second dose of the day (doses two and four) than the corresponding first dose (doses one and three).

For subject 2, the behavioral baseline in the OFF drug state included consistent communication and command-following, and the CRS-R was at ceiling. As documented by structured clinical observations following nine doses, this patient's zolpidem-induced behavioral facilitation included marked improvements in oro-motor control of chewing and swallowing, and increased verbal fluency. Additionally, there was a suppression of a coarse tremor and restoration of goal-directed fine motor-control of the dominant right arm and hand. This subject had received three daily doses of zolpidem for five years at the time of the study; an earlier report of zolpidem induced behavioral facilitation in this subject from the third year of use of the medication (*Clauss and Nel, 2006*) indicated a shift from a level II Rancho Los Amigos Cognitive Score to a level VI-VII indicating that at baseline purposeful vocalization was not observed in the OFF drug state at that time.

For subject 3, the behavioral baseline was characterized by reproducible responses to command, visual fixation, automatic motor responses, intelligible verbalization and intentional communication. These behavioral responses, however, could only be elicited with vigorous arousal enhancing maneuvers. After zolpidem administration on a single formal assessment, CRS-R score increased from 15 to 16. This one-point change reflected a change in the CRS-R arousal subscale score from 1 to 2: a shift from a state that required constant stimulation to maintain arousal to a state characterized by spontaneous eye-opening and behavioral responsiveness. Other CRS-R subscale scores did not change but no longer depended on external stimulation to elicit responsiveness. In addition, the subject also showed increased alertness (as judged by orientation to sensory stimuli) and agitated movements, with an increase in speech rate and words that were subjectively rated as more intelligible (see 'Methods' Clinical Histories).

## Effects of zolpidem administration on the EEG power spectrum

To characterize changes in spontaneous brain dynamics associated with zolpidem administration, we analyzed the EEG around the times of zolpidem administration. Subjects 1 and 2 were studied over several days in a hospital setting; Subject 3's evaluation was limited to assessment of a single dose of zolpidem while at home ('Methods'). We begin by comparing the EEG in the hour preceding and following zolpidem administration, and then consider its further evolution in time.

*Figure 2* shows results from the frontal midline electrode pair Fz-Cz for all three subjects. Prior to zolpidem administration, each subject's data demonstrates a peak of power in the range 6–10 Hz. In all cases, this peak is sharply attenuated 1 hr following drug administration, and there is an increase

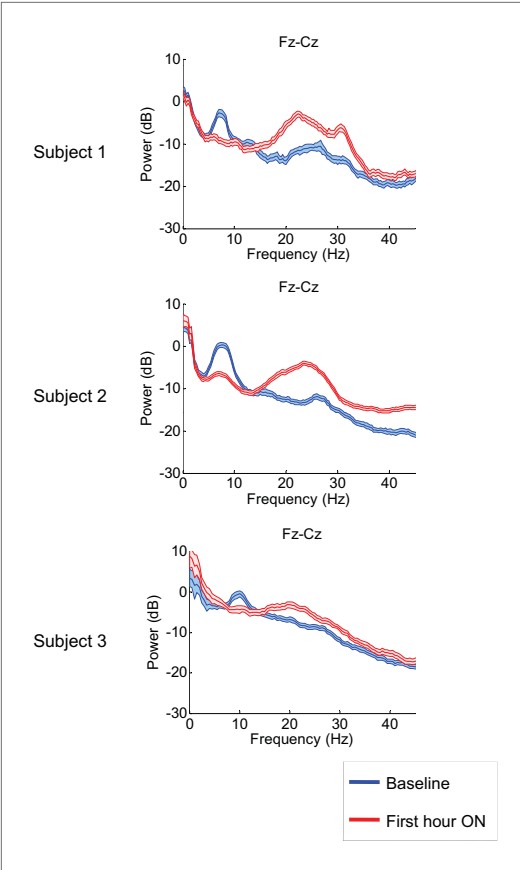

**Figure 2**. Power spectra estimated from midline EEG channel Fz-Cz recordings. Power spectra from all three subjects (mean and 95% confidence intervals). Red: average spectral power in the hour prior to each zolpidem dose. Blue: 20–60 min after the zolpidem dose. Narrow, low frequency spectral peaks are apparent in the pre-drug state that are attenuated in the hour post-dose. Beta range peaks (20–30 Hz) are apparent in all post-drug spectra. Changes in ~6–10 Hz peak between conditions are significant by two-group test (see 'Methods'; *Table 1*).

The following figure supplements are available for figure 2:

**Figure supplement 1**. Analysis of EEG segments obtained during ON and OFF drug periods from Subject 1's second zolpidem administration (cf. *Figure 1*; *Table 1*) subdivided according to periods of low and high levels of environmental stimulation, a distinction based on detailed review of the simultaneously recorded video.

in power in the range ~15–30 Hz (p<0.05 by a z-statistic, the Two Group Test (*Bokil et al., 2007*), see 'Methods' for approach to significance testing, *Figure 3—figure supplement 1*; *Tables 1 and 2*). *Figure 2—figure supplement 1* shows that the changes in spectral shape do not represent a 'performance confound'. Specifically, we determined EEG power spectra only from periods in which the patient was at rest and not interactive with persons or objects in their environment and compared this with the power spectra obtained during active interactions with people at the bedside. The level of environmental stimulation did not influence the spectrum, and the zolpidem effect of *Figure 2* was similarly present for both periods of either low or high levels of environmental stimulation.

*Figure 3*, *Figure 3—figure supplement 1*; *Tables 1 and 2* show that in each patient, the changes in spectral shape following zolpidem administration are widely distributed across the electrode array.

Although some individual channels show additional specific features (e.g., a low frequency peak at ~12 Hz in channel pair C4-CP2 in Subject 1), the low frequency oscillations ~6–10 Hz are diffusely present across brain regions in the baseline condition in all subjects, and absent following zolpidem. Similarly, widely-distributed increases in ~15–30 Hz power are observed after zolpidem administration in all three subjects (*Figure 3*).

Analysis of responses to multiple drug administrations was carried out in Subject 1 (four administrations) and Subject 2 (five administrations). *Figure 4* shows the spectra at Fz-Cz in order of decreasing washout time of the prior zolpidem dose ( *Tables 1 and 2*). Consistent with the data shown in *Figures 2 and 3*, all transitions show a decrease in power at 6–10 Hz and an increase from 15–30 Hz following drug administration (p<0.05). In both subjects, the low-frequency peak is least prominent when prior washout period is shortest, suggesting that there is a carry-over effect at these dosing intervals.

Further analysis of changes in the power spectrum across different washout intervals after each dose revealed three additional findings: (1) a clear distinction between changes in resting low frequency oscillations present over anterior (frontocentral-temporal) and posterior (centro-parietal-occipitotemporal) channels, (2) a common average resting low frequency peak across anterior channels, and (3) a consistent increase in the average center frequency of the posterior but not the anterior EEG channels with shorter dosing intervals. Comparison of the anterior and posterior channels in both subjects reveal a consistent finding of lower resting average low frequency in anterior channels typically ~7.4 Hz. In *Table 1* , the average center frequency for anterior EEG channels in Subject 1 ranges from 7.1 Hz to 7.4 Hz, while posterior EEG channels show a higher average center frequency ranging from 7.5 Hz to 8.3 Hz. A sharper

**Table 1.** Relationship of low frequency peak in OFF zolpidem baseline and time interval between doses

**SUBJECT 1 center frequency for low frequency peak in hour prior to dose across channels and transitions**

| Channel | Pairs | Baseline 1 (62 hr off drug) 9:30AM, 1st | Significant On vs off suppression | Baseline 2 (17 hr off drug) 12:00PM, 1st | Significant On vs off suppression | Baseline 3 (9 hr off drug) 6:30PM, 2nd | Significant On vs off suppression | Baseline 4 (5 hr off drug) 5:00PM, 2nd | Significant On vs off suppression |
|---|---|---|---|---|---|---|---|---|---|
| Fpz | Fp1 | 7.7 | * | 7.3 | * | 7.7 | * | – | – |
| Fpz | Fp2 | 7 | * | 7.3 | * | 7.3 | * | – | – |
| Fp1 | F3 | 7.7 | * | 7.3 | * | 7.3 | * | 7 | * |
| Fp2 | F4 | 7.3 | * | 7.3 | * | 7.3 | * | 6.3 | * |
| F1 | FC1 | 7.3 | * | 7.7 | * | 7.3 | * | 7 | * |
| FC1 | C3 | 7.3 | * | 7.7 | * | 7 | * | 7 | * |
| FC2 | C4 | – | – | 7.3 | * | 7 | * | – | – |
| F3 | FC1 | 6.7 | * | 7.3 | * | 6.7 | * | 7.3 | * |
| F4 | FC2 | 7.3 | * | 7.7 | * | 7 | * | 7.7 | * |
| F2 | FC2 | 7.3 | n.s. | 7.7 | * | 7 | * | 7 | n.s. |
| Fp1 | AF7 | 6.3 | * | 7.7 | * | 7.3 | * | 7 | * |
| Fp2 | AF8 | 7.3 | * | 7.3 | * | 7.3 | * | – | – |
| AF7 | F7 | 6.7 | * | 7.3 | * | 7.3 | * | 6.3 | * |
| AF8 | F8 | 7 | * | 7.3 | * | 7.3 | * | – | – |
| F7 | FC5 | 6.7 | * | 7.3 | * | 7.3 | * | 6.3 | * |
| F8 | FC6 | 7.7 | * | 7.3 | * | 7.3 | * | 8.3 | n.s. |
| FC5 | T3 | 6 | * | 7.3 | * | 7.3 | * | 7.3 | * |
| FC6 | T4 | 7.3 | n.s. | 7.3 | * | – | – | – | – |
| T3 | CP5 | – | – | 7.3 | * | 7.3 | * | 9.7 | * |
| T4 | CP6 | 7.7 | * | 7.3 | * | 7.7 | n.s. | 9.3 | * |
| C3 | CP1 | – | – | 7.7 | * | 7 | * | 7 | * |
| C4 | CP2 | – | – | 7.7 | * | 7.3 | * | 7 | * |
| Fz | Cz | 7.3 | * | 7.7 | * | 7 | * | 7 | * |
| Anterior Channels | | | | | | | | | |
| Average Center Freq | | 7.1 | | 7.4 | | 7.2 | | 7.3 | |
| Standard Deviation | | 0.5 | | 0.2 | | 0.2 | | 1.0 | |
| T5 | PO7 | – | – | 7.3 | * | 7.3 | * | 6.7 | n.s. |
| T6 | PO8 | 7.7 | * | 7.3 | * | 7.7 | * | 11 | * |
| CP1 | P3 | 7.7 | * | 7.7 | * | 6.7 | * | 6.7 | * |
| CP6 | T6 | 7.7 | * | 8.3 | * | – | – | 8 | n.s. |
| CP2 | P4 | 7.3 | n.s. | 8 | * | – | – | 9 | n.s. |
| CP5 | T5 | – | – | 7.7 | * | 7.3 | * | 9.7 | * |
| P3 | O1 | 7.3 | * | 7.7 | * | 7.3 | * | 9 | * |
| P4 | O2 | – | – | 8 | * | – | - | – | – |
| PO7 | O1 | 7.3 | * | 7.3 | * | 7.7 | * | 9 | * |
| PO8 | O2 | 7.7 | * | 7.7 | * | 7.3 | * | - | – |
| Cz | Pz | – | – | 7.7 | * | 7 | * | 7 | * |

*Table 1. Continued on next page*

*Table 1. Continued*

**SUBJECT 1 center frequency for low frequency peak in hour prior to dose across channels and transitions**

| Channel | Pairs | Baseline 1 (62 hr off drug) 9:30AM, 1st | Significant On vs off suppression | Baseline 2 (17 hr off drug) 12:00PM, 1st | Significant On vs off suppression | Baseline 3 (9 hr off drug) 6:30PM, 2nd | Significant On vs off suppression | Baseline 4 (5 hr off drug) 5:00PM, 2nd | Significant On vs off suppression |
|---|---|---|---|---|---|---|---|---|---|
| CPz | POz | 7.3 | n.s. | – | – | 6.7 | * | 6.7 | * |
| POz | Oz | 7.7 | * | 7.3 | * | 7.3 | * | – | – |
| Posterior Channels | | | | | | | | | |
| Average Center Freq | | 7.5 | | 7.7 | | 7.2 | | 8.3 | |
| Standard Deviation | | 0.2 | | 0.3 | | 0.3 | | 1.5 | |
| Average across all | | | | | | | | | |
| Average Center Freq | | 7.3 | | 7.5 | | 7.2 | | 7.7 | |
| Standard Deviation | | 0.4 | | 0.3 | | 0.3 | | 1.2 | |

Table 1 shows average center frequency of low frequency peak at~6–10 Hz, if present, across all channels for Subjects 1 in relation to time off zolpidem prior to 1 hr baseline EEG measurements and clock time of each dose. In both patients, shorter zolpidem dosing intervals are associated with higher center frequencies across posterior but not anterior EEG channels. Similarly, an increased standard deviation of the measurement is observed as interval between doses is shorter for the posterior EEG channel measurements. Of note, for both subjects, anterior EEG channel pre-zolpidem dose baselines revealed a consistent~7.4 Hz average center frequency with a small standard deviation. The consistency of these findings despite the wide difference in their underlying etiologies of injury support the proposed common cellular and circuit mechanism; the correspondence of the~7.4 Hz peak and intrinsic oscillation frequency of neocortical Layer V cells (*Silva et al., 1991*) also support this model.

difference for anterior and posterior EEG channels is seen in *Table 2* for Subject 2; for this subject, anterior EEG channels show an average center frequency of ~7.4 Hz for four of the five measured baselines , with the notable exception of the second baseline measurement. The second baseline measurement in Subject 2 occurred early in morning (8:30AM) and may reflect an independent influence of arousal state, as a similar effect of increased baseline 15–30 Hz power is seen for this pre-dose baseline (refer to *Figure 4* and discussion below). Posterior EEG channels in Subject 2 show an average center frequency range from 8.0 Hz to 9.4 Hz.

*Tables 1 and 2* show, in order of decreasing dosing (washout) interval, the average center frequency of low frequency peaks in the baseline spectral power across EEG channels for Subjects 1 and 2 prior to each administration of zolpidem. For the longest washout intervals compared to the shortest, the baseline spectral peaks appear at lower frequencies across all channels for both Subjects 1 and 2 (*Tables 1 and 2*). For the shortest washout intervals (*Tables 1 and 2*), however, the posterior channels demonstrate a persistent shift in the average center frequency suggesting a carry-over effect for these regions of the brain. For example, in the baseline prior to dose four for Subject 1 several EEG electrode derivations show an increase in the baseline low frequency peak including an ~11 Hz peak arising posteriorly in channel pair T6–PO8; for Subject 2 an 11 Hz peak is similarly seen in their 4 and 5 hr washout baselines in the T5–O1,T6–O2 and T3–T5 channel pairs.

There are other differences between the spectral changes during the individual drug administrations, but their interpretation is unclear because of the limited number of drug administrations that we were able to study. In some instances increased power in the ~15–30 Hz range is also apparent before and after the drug administration. This variation is substantial and suggests a fluctuation in the baseline state possibly linked to differences in arousal state or other factors not controlled for in this observational study. For Subject 2, a marked change in the power spectrum in the ~15–30 Hz range is limited primarily to the first dose which followed a 20 hr washout period and was received late in the afternoon (4:30 PM), suggesting that the regular dosing schedule in this subject of three

**Table 2.** Relationship of low frequency peak in OFF zolpidem baseline and time interval between doses

### SUBJECT 2 center frequency for low frequency peak in hour prior to dose across channels and transitions

| Channel | Pairs | Baseline 1 (20 hr off drug) 4:30PM, 1st | Significant On vs off suppression | Baseline 2 (16 hr off drug) 8:30AM, 1st | Significant On vs off suppression | Baseline 3 (6 hr off drug) 4:30PM, 2nd | Significant On vs off suppression | Baseline 4 (5 hr off drug) 5:00PM, 3rd | Significant On vs off suppression | Baseline 5 (4 hr off drug) 12:15PM, 2nd | Significant On vs off suppression |
|---|---|---|---|---|---|---|---|---|---|---|---|
| Fp1 | F3 | 7 | * | – | n.s. | 7.3 | * | 7.7 | n.s. | 8 | n.s. |
| Fp2 | F4 | 7 | * | 8.5 | * | 7.3 | * | 7 | * | 7.7 | n.s. |
| F3 | C3 | 7.7 | * | 8.5 | * | 7.7 | * | 7.7 | * | 7.7 | * |
| F4 | C4 | 7.7 | * | 8 | * | 7.7 | * | 7.7 | * | 7.3 | * |
| F7 | T3 | 7.7 | n.s | – | * | 7.7 | n.s | 7.7 | n.s. | 6 | n.s. |
| F8 | T4 | (No data) | | 8 | n.s. | 7.3 | * | 7 | n.s. | 7.3 | n.s. |
| Fp1 | F7 | 7 | n.s | – | * | 7.7 | * | 7.7 | n.s. | 7 | n.s. |
| Fp2 | F8 | 7.7 | * | – | n.s. | 7 | * | 6.7 | n.s. | 7.3 | n.s. |
| Fz | Cz | 7.7 | * | 8.5 | * | 7 | * | 7.7 | * | 7 | n.s. |
| **Anterior Channels** | | | | | | | | | | | |
| Average Center Freq | | 7.4 | | 8.3 | | 7.4 | | 7.4 | | 7.3 | |
| Standard Deviation | | 0.4 | | 0.3 | | 0.3 | | 0.4 | | 0.6 | |
| C3 | P3 | 8.3 | * | 8.5 | * | 7.7 | * | 8.3 | * | 8.7 | n.s. |
| C4 | P4 | 8 | * | 9 | * | 9.3 | * | 8.3 | * | 7.7 | n.s. |
| P3 | O1 | 7.7 | * | 9 | * | 11.3 | * | 8.7 | * | 8 | n.s. |
| P4 | O2 | 8.3 | * | 9 | * | 10 | * | 8.7 | * | 8.3 | n.s. |
| T3 | T5 | 7 | * | 8.5 | * | 7.7 | * | 11.7 | n.s. | 11.7 | n.s. |
| T4 | T6 | (No data) | | 9 | * | 9.3 | n.s | 10 | * | 10 | n.s. |
| T5 | O1 | 8 | * | 9 | * | 10 | * | 8.3 | * | 11.3 | n.s. |
| T6 | O2 | 8 | * | 9.5 | * | 10 | * | 9.7 | * | 11 | n.s. |
| Cz | Pz | 8.3 | * | 8.5 | * | 9 | * | 8.7 | * | 7.7 | n.s. |
| **Posterior Channels** | | | | | | | | | | | |
| Average Center Freq | | 8.0 | | 8.9 | | 9.4 | | 9.2 | | 9.4 | |
| Standard Deviation | | 0.4 | | 0.3 | | 1.2 | | 1.1 | | 1.6 | |
| Average Center Freq | | 7.7 | | 8.7 | | 8.4 | | 8.3 | | 8.3 | |
| Standard Deviation | | 0.5 | | 0.4 | | 1.3 | | 1.2 | | 1.6 | |

Table 2 shows average center frequency of low frequency peak at~6–10 Hz, if present, across all channels for Subjects 2 in relation to time off zolpidem prior to 1 hr baseline EEG measurements and clock time of each dose. In both patients, shorter zolpidem dosing intervals are associated with higher center frequencies across posterior but not anterior EEG channels. Similarly, an increased standard deviation of the measurement is observed as interval between doses is shorten for the posterior EEG channel measurements. Of note, for both subjects, anterior EEG channel pre-zolpidem dose baselines revealed a consistent~7.4 Hz average center frequency with a small standard deviation. The consistency of these findings despite the wide difference in their underlying etiologies of injury support the proposed common cellular and circuit mechanism; the correspondence of the~7.4 Hz peak and intrinsic oscillation frequency of neocortical Layer V cells (*Silva et al., 1991*) also support this model. Of note as an outlier is the second baseline for this subject which is the only measurement early in the morning (8:30) suggesting a possible interaction with diurnal activity of the brain arousal system, however, insufficient data is available to establish this linkage.

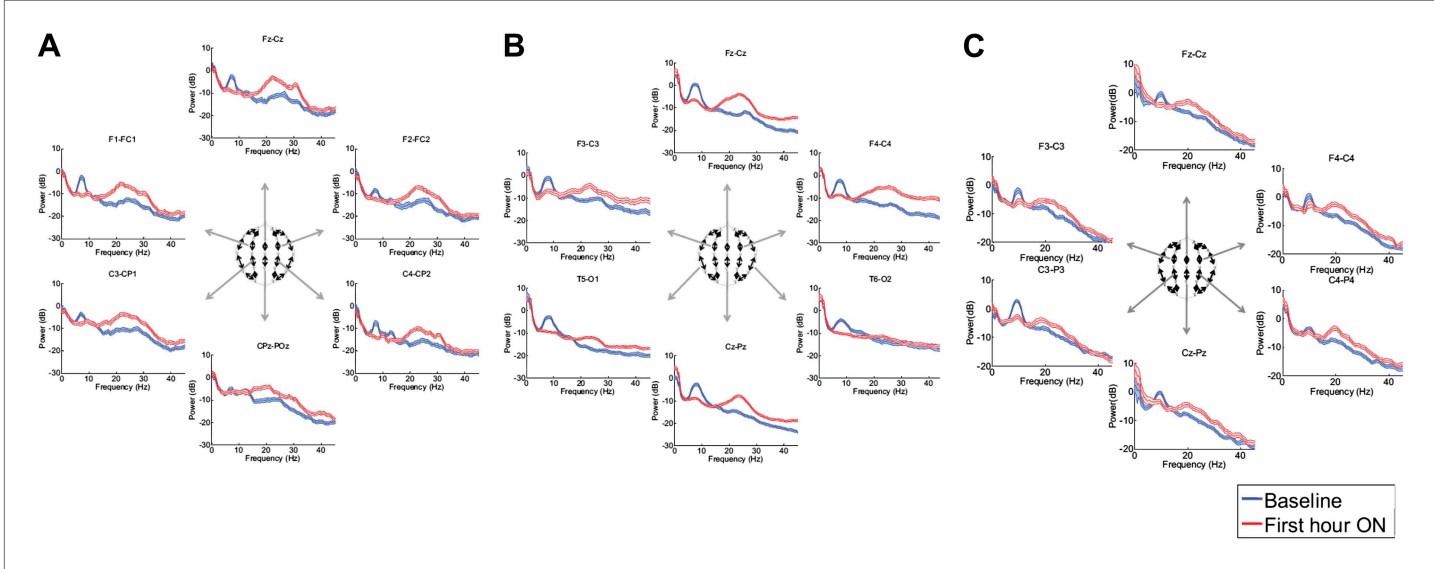

**Figure 3**. Power spectra (mean and 95% confidence intervals) estimated from selected EEG channel recordings across the head from all three subjects (**A**, **B**, and **C**). Red: average spectral power in the hour prior to each zolpidem dose. Blue: 20–60 min after the zolpidem dose. Narrow, low frequency spectral peaks are apparent in the pre-drug state that are attenuated in the hour post-dose. Beta range peaks (~20–30 Hz) are apparent in all post-drug spectra. Changes in ~6–10 Hz peak between conditions are significant by two-group test ('Methods'; **Table 1**).

The following figure supplements are available for figure 3:

**Figure supplement 1**. Significance testing of power across frequencies comparing the OFF and ON zolpidem states for Subject 1, transition 1 (baseline vs first hour) is summarized using two-group test (see 'Methods'; **Bokil et al., 2007** for further methods).

doses per day may produce carry-over effects that diminish once the subject is off drug for more extended time periods. In addition to diurnal arousal effects, carry-over effects also appear to affect the 15–30 Hz component as a broad ~15–30 Hz peak also remains present in this range for the shortest washout baselines in Subject 1 (4 hr; **Figure 4D**). The main findings of suppression of ~6–10 Hz activity and increase of ~15–30 Hz activity with zolpidem administration showed a general robustness across all three subjects and in Subjects 1 and 2 across separate assessments of ON and OFF drug doses (**Tables 1 and 2**).

Consistent patterns of EEG dynamics across Subjects 1 and 2 are also observed over the 2.5–3 hr course of each dose, as shown in **Figure 5**. In both subjects, the increase in high frequency power has two phases: maximal increases in power ~15–30 Hz occur within the first 30–40 min, and this is followed by an attenuation of power and narrowing of frequency range to ~20–30 Hz. By 2 hr after dosing, the enhanced high-frequency power has noticeably decayed. In contrast, both subjects show suppression of low-frequency power beyond this point (up to 3 hr in subject 1, and up to 4 hr in subject 2), suggesting that the prolonged effect on low-frequency activity may underlie the 'carry-over' effect. (Data were not collected for subject 3 beyond the first hour after zolpidem administration and similar analyses are thus not available).

Of particular interest is a finding observed in two posterior channels from subject 1 in one of the three transitions analyzed (dose two). As seen in the third spectrogram (bottom panel) shown in **Figure 5A**, the posterior channels CPz-POz reveal the appearance of a ~10 Hz feature in the spectrogram around 1.25–1.5 hr after administration of the dose (green arrow). This feature of the CPz-POz spectrogram is distinct from the low frequency ~7 Hz feature (red arrows) and high frequency 15–25 Hz feature (white arrows) seen in the other EEG channel spectrograms (**Figure 5A**). While not reproduced in other transitions, this feature suggests a possible restoration of the posterior alpha rhythm arising later in the course of ON drug state.

## Effects of zolpidem on EEG coherence

To determine whether zolpidem administration altered the relationship between activity across cortical regions, we examined its influence on EEG coherence. Representative findings are shown in **Figure 6A,B,C**

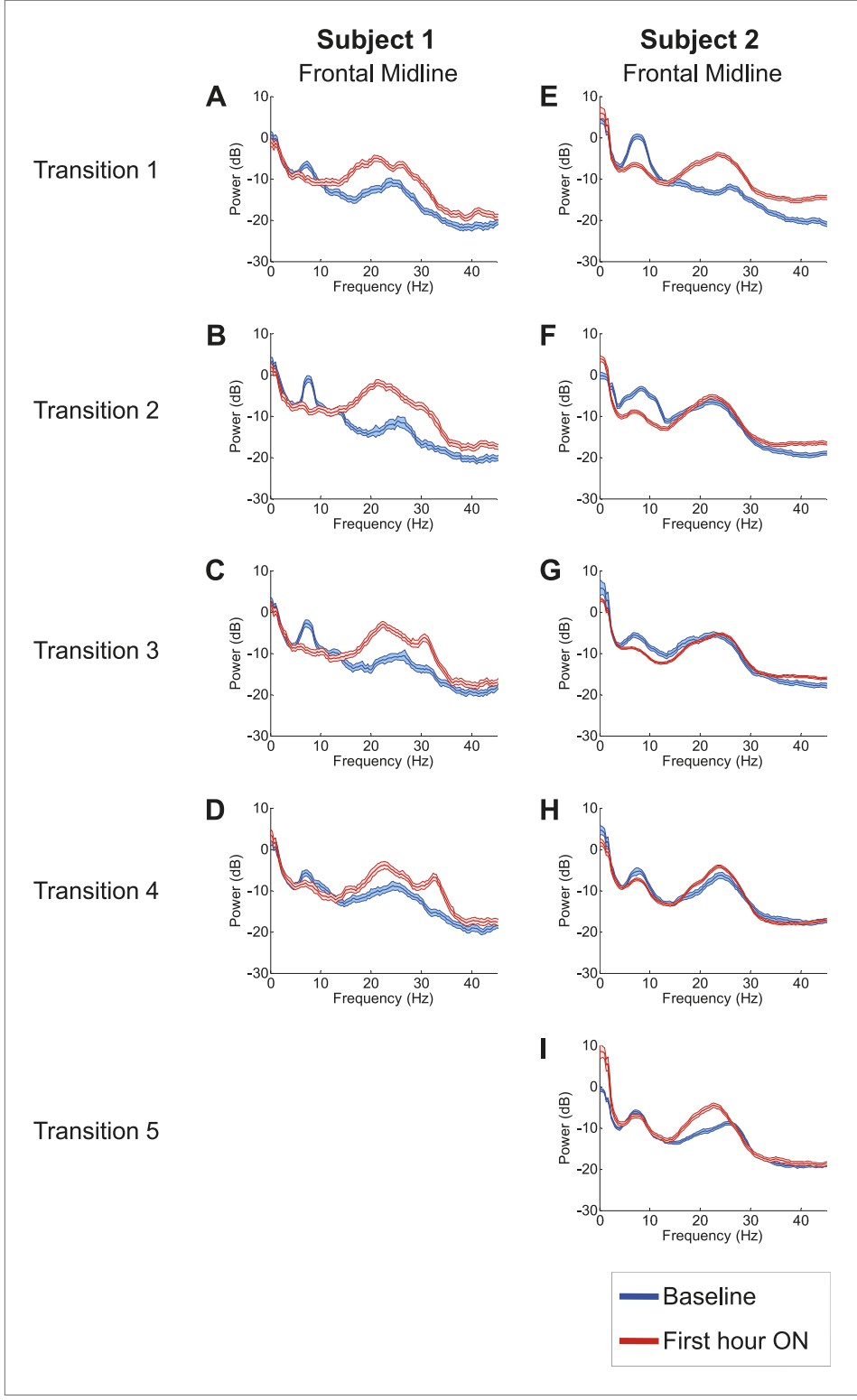

**Figure 4**. Power spectra from midline channels for subjects 1 and 2 across multiple transitions from a washout baseline to ON drug state. Eyes open, awake epochs selected from one hour prior to 1 hr subsequent to each dose (Red) are shown and compared to 20–60 min after the zolpidem dose (Blue). Changes in ~6–10 Hz peak between conditions are significant by two-group test (*Tables 1 and 2*).

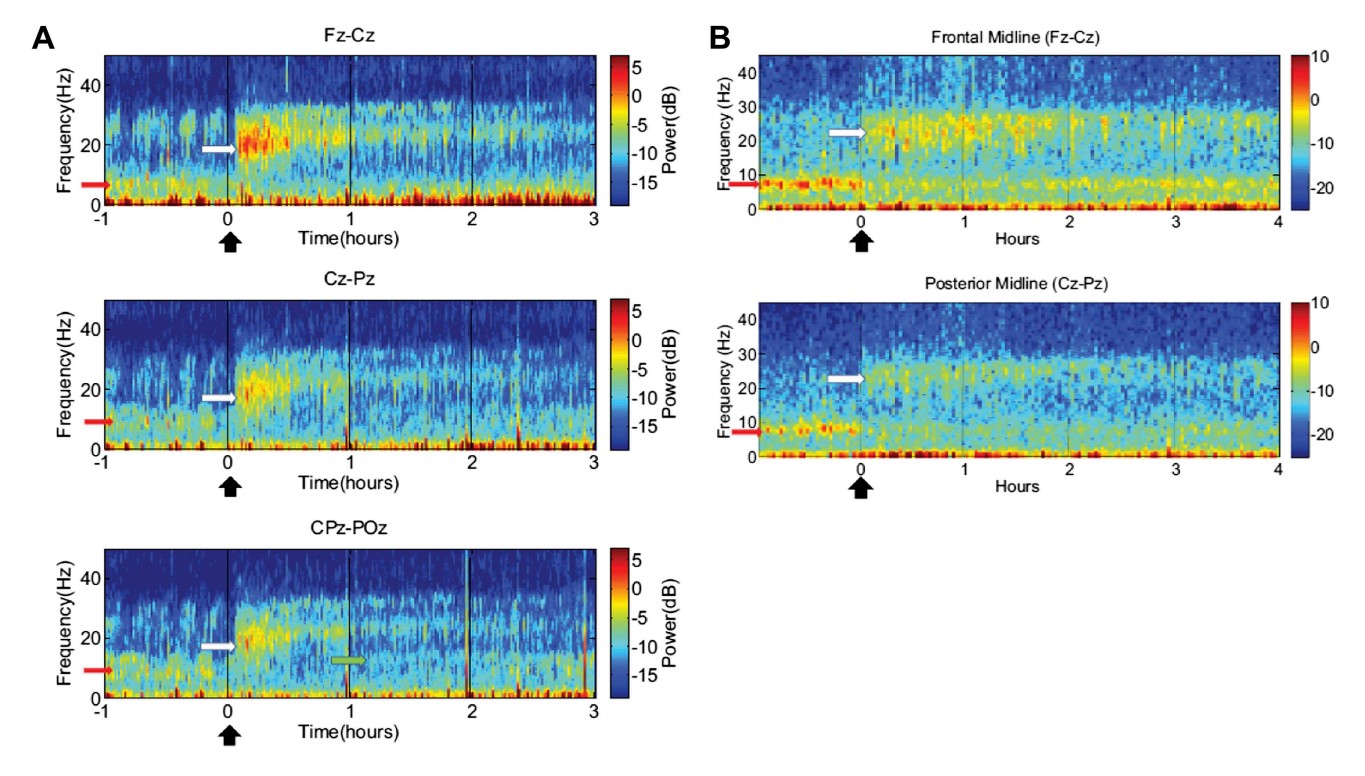

**Figure 5**. Time-frequency analysis of selected EEG channels from Subject 1 (**A**) and Subject 2 (**B**). Frontal and posterior midline channels, for both subjects demonstrate a low frequency peak in power (red arrows) during the hour prior to zolpidem administration attenuates within the first 10–15 min after the drug is given. This corresponds to the time period when the subject begins to manifest improved behavioral function. Concomitant with the attenuation of the low frequency peak, there appears a broader ~15–25 Hz peak during the 30 min after the drug is given (white arrows), which narrows and reduces attenuates slowly over the next 2–3 hr. In Subject 1 (**A**), a ~10 Hz peak appears approximately 1 hr into the post-dose period in CPz-POz channel (green arrow).

(and *Figure 6—figure supplement 1*) for intra-hemispheric coherence, and in *Figure 7* for inter-hemispheric coherence(and *Figure 7—figure supplement 1*). Consistently across subjects, the OFF drug state was associated with a coherence peak range 0.6–0.8 at ~6–10 Hz , the same range as the low-frequency power peak (p<0.05, two-group test). Zolpidem administration reduced the level of coherence for most channels to the range 0.3–0.5. In subjects 1 and 2, this was typically accompanied by an increase in coherence in the range of the high-frequency power peak (20–30 Hz). These changes are most prominent in the transitions following the longest washout periods for Subjects 1 and 2 (Doses one to three).

## Discussion

Paradoxical behavioral facilitation in response to zolpidem in severely brain-injured subjects with disorders of consciousness is well-documented, but relatively rare, with only one of 15 subjects identified in a prospective study (*Whyte and Myers, 2009*). Here we studied three such patients with disparate etiologies of brain injury, who had very different underlying patterns of structural brain injury and mechanisms of widespread neuronal death and disconnection. Despite these differences, quantitative EEG analysis revealed strikingly consistent dynamical signatures of brain dynamics in the resting awake state, and how these dynamics were affected by zolpidem. In all subjects, resting EEG power spectra in the baseline state revealed an abnormal peak at ~6–10 Hz and high coherence in this same frequency range across intra- and inter-hemispheric brain regions. Behavioral activation with zolpidem reliably attenuated ~6–10 Hz power and coherence in all subjects and associated with an increase in power at ~15–30 Hz. These findings suggest a fundamental similarity in the dynamics of brain states that are poised to elicit a zolpidem response. Below we propose a physiological basis for these common resting

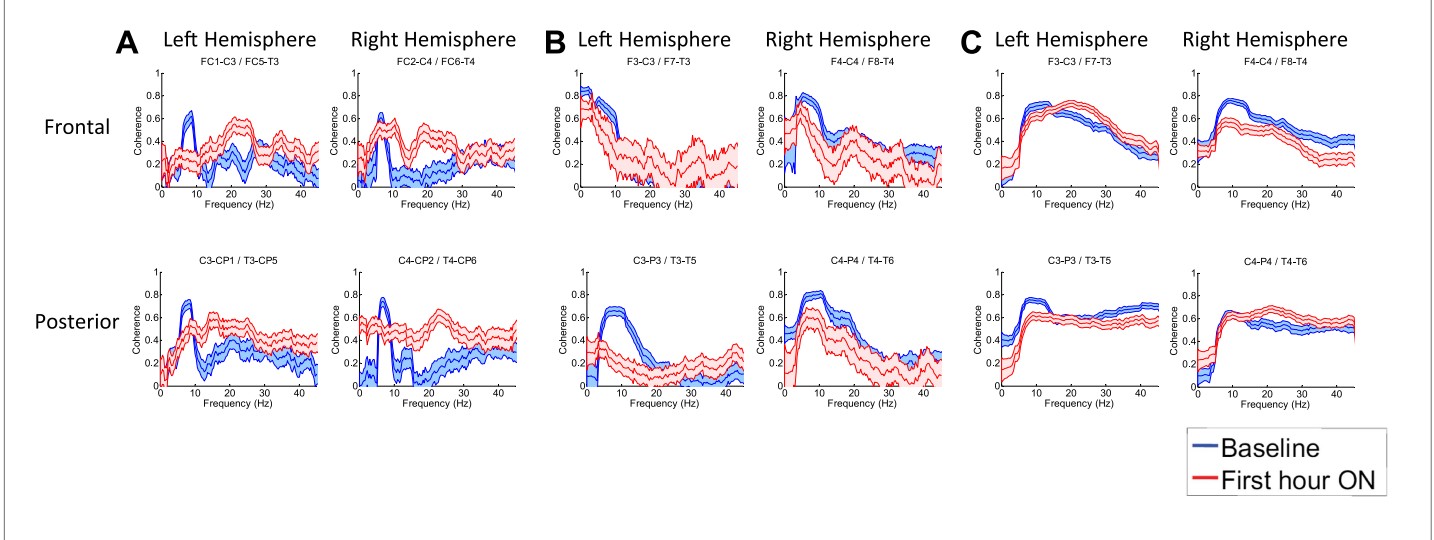

**Figure 6**. (**A**). Intra-hemispheric coherences (subject 1). Pre-drug coherence peaks at 6–10 Hz. This peak is attenuated in the hour after the drug is given. Changes in ~6–10 Hz peak between conditions are significant by two-group test ('Methods'; *Figure 6 Figure Supplement 1*). (**B**) (subject 2). Intra-hemispheric coherences (subject 2). Pre-drug coherence peaks are evident at ~ 6–10 Hz. This peak is attenuated in the hour after the drug is given. (**C**) (subject 3). Intra-hemispheric coherences (subject 3). Pre-drug coherence peaks at ~ 6–10 Hz. This peak is attenuated in the hour after the drug is given.

The following figure supplements are available for figure 6:

**Figure supplement 1**. Significance testing of intra-hemispheric coherences for Subject 1, transition 2 (baseline vs first hour) is summarized using two-group test (see 'Methods'; *Bokil et al., 2007* for further methods).

state dynamics identified by our EEG findings in the OFF drug state that show specific correspondence with known physiological processes. We further discuss a key prediction of this model: that changes observed in EEG spectral content during activation in the ON drug state will covary with the degree of structural and functional deafferentation of the cerebral cortex.

It is important to note that we are stopping short of proposing that a 6–10 Hz spectral peak with high spatial coherence is a predictor of zolpidem responsiveness. We suspect that the apparent rarity of the zolpidem response is not due to the incidence of this EEG feature (which is unknown), but to other factors, such as the 50-fold variations of GABA receptor subtype expression in the human population (*Kang et al., 2011*). Testing these ideas, however, goes far beyond the scope of the present study, as it would require a large structured, prospective, randomized clinical trial (*Whyte and Myers, 2009*). We argue below, however, that these EEG findings may be a general marker of a recruitable reserve that in principle may be recruited by other neuromodulatory interventions (*Schiff, 2010*), and not just zolpidem.

### Proposed cellular mechanisms underlying broadly coherent, low frequency oscillations observed in the OFF zolpidem state

As a result of diverse mechanisms of injury, all three severely brain-injured subjects suffered widespread neuronal death and consequent deafferentation of many neurons across the cerebral cortex and subcortical structures. The implications of severe deafferentation of cortical neurons for EEG dynamics can be estimated by comparing two extremes: completely de-afferented 'slabs' of cortical tissue, and normal cortical tissue. In the former case (in vivo studies of completely deafferented 'slabs'), neocortical neurons typically have an intracellular potential of ~-70 mV and the dominant frequency of activity measured in far field potentials is ~1 Hz (*Timofeev et al., 2000*). Intact, fully connected neocortical neurons have an average membrane potential in active wakeful states, that ranges from ~−65 to −55 mV (*Steriade, 2001*) and the dominant frequency of far-field potentials is in the 8–12 Hz range. Given these benchmarks it is reasonable to assume that in our patient subjects, all with very significant but incomplete deafferentation, that average membrane potentials are intermediate between these conditions. In this context, the prominent low frequency peaks seen across

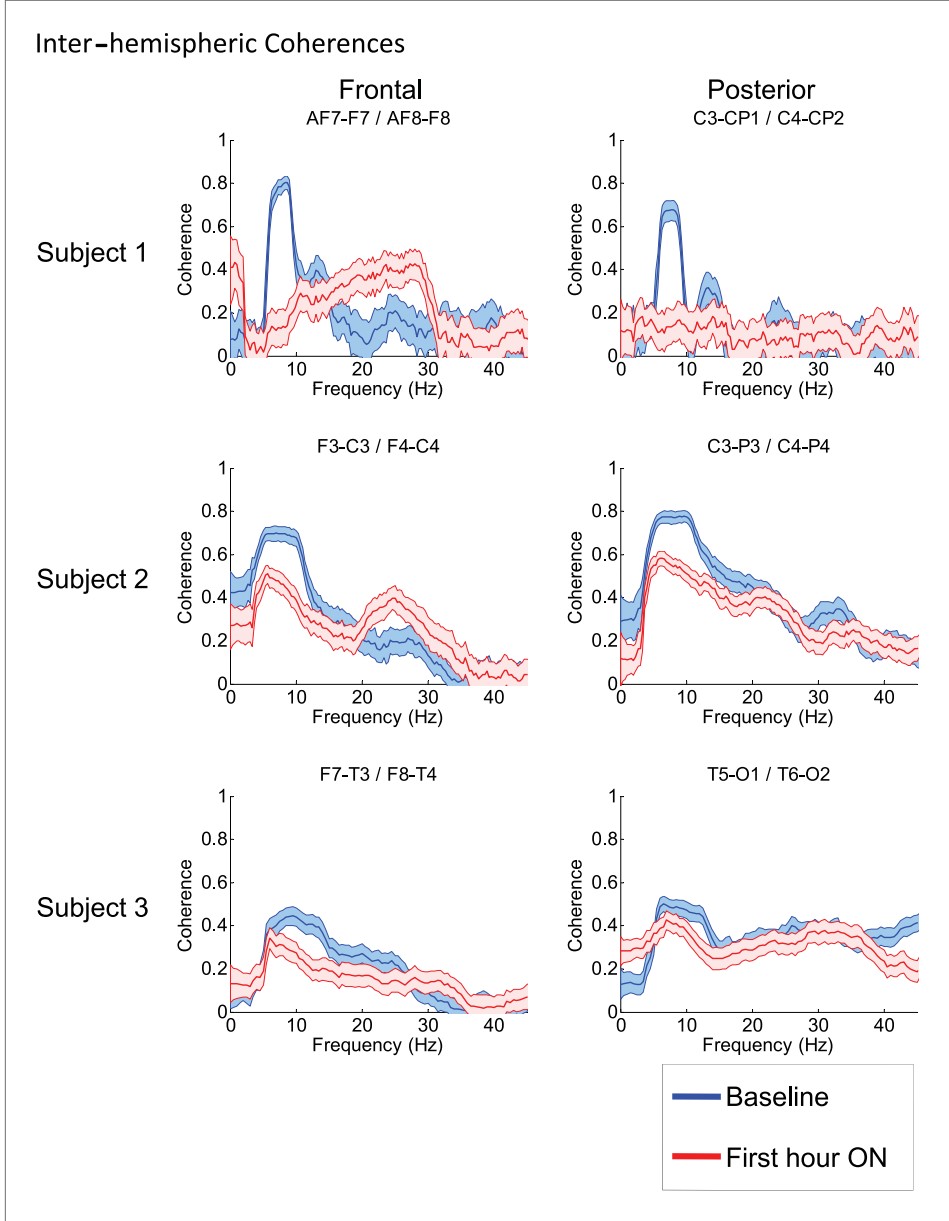

**Figure 7**. Inter-hemispheric coherences all three subjects. Pre-drug coherence peaks at 6–10 Hz are seen in frontal and posterior channels comparisons for all subjects. This peak is attenuated in the hour after the drug is given. Changes in ~6–10 Hz peak between conditions are significant by two-group test (see 'Methods'; **Figure 7— figure supplement 1**, for Subject 1 coherences for example of coherence changes across all channel pairs in single dose).

The following figure supplements are available for figure 7:

**Figure supplement 1**. Significance testing of inter-hemispheric coherences for Subject 1, transition 2 (baseline vs first hour) is summarized using two-group test (see 'Methods'; **Bokil et al., 2007** for further methods).

the power spectra from anterior EEG channels and across different OFF drug baselines of ~7.4 Hz are notable for comparison to findings from in vitro studies of spontaneous oscillations generated by neocortical pyramidal cells. Silva et al. (**Silva et al., 1991**) identified that nearly identical frequencies of sustained firing lasting over seconds (7.4 Hz SD +/−0.6) are produced by layer V neocortical pyramidal cells in response to very brief (4 ms) current pulses. These sustained oscillations could be initiated under physiologic conditions in vitro that maintain intrinsic membrane potentials near firing threshold

at ~−60–65 mV (*Silva et al., 1991*). As seen in *Tables 1 and 2*, an average center frequency of 7.4 Hz (~SD +/−0.6) characterizes all but one of nine separately measured baselines across the anterior EEG channels of Subjects 1 and 2.

Thus, we propose that the markedly deafferented conditions of cortical neurons in the reduced slice preparation may reasonably approximate the markedly deafferented state of the cortex in our three subjects and that self-sustained intrinsic membrane oscillations arising from Layer V pyramidal neurons triggered by random synaptic background activity are the underlying mechanism for the origin of the ~6–10 Hz activity present in the resting EEG at baseline. In the slice preparation, increases in membrane depolarization with stronger and more sustained injected currents produced shifts of these intrinsically generated oscillatory firing patterns toward higher frequencies (up to 12 Hz) when kept below firing threshold (*Silva et al., 1991*) indicating that a wider range of frequencies beyond 7.4 Hz could in principle still reflect such pathologically driven oscillations (and in Subject 3 the resting oscillations tend to be at higher frequencies as shown in *Figures 1 and 3C*). Moreover, fully depolarizing these neurons in the slice results in similar shifts to ~15–30 Hz firing rates (*Silva et al., 1991*).

If the intrinsic properties of partially de-afferented cortical tissue accounts for the peak of EEG activity in the 6–10 Hz range, then what could account for its high levels of spatial coherence? We propose that this coherence arises from a simple straightforward dynamical mechanism: the 'Huygens' clock principle' (refer to *Huygens, 1665*; *Rosenblum and Pikovsky, 2003*). According to this principle, when oscillators with similar frequencies interact weakly, stable synchronization around a consensus frequency is the generic result. Here, we hypothesize that the net effects of residual Layer V output neurons (efferent to other cortical regions either directly or via subcortical structures) constitute a weak coupling. The predicted result is widespread coherence of cortical columns at a consensus theta frequency (~7.4 Hz), as we in fact observe in all three subjects.

We speculate that our model for common baseline EEG dynamics may also account for other widespread rhythmic oscillations that are described in the cardiac arrest and other types of severe anoxic brain injury. These oscillations are in the alpha (*Young et al., 1994*) and less commonly theta (3–7 Hz) frequency range. Slower frequencies and lack of a change in frequency in response to sensory stimulation are associated with worse outcomes (*Young et al., 1994*), consistent with the notion that this activity results from de-afferentation, and the dominant frequency range indexes its severity.

However, we do not propose that all forms of pathological rhythmic activity arise from this mechanism, only to those in which the structural brain injuries are severe and diffuse. Theta (3–7 Hz) activity has also been reported in patients with focal cortical abnormalities or neuropsychiatric disorders without widespread neuronal loss or disconnection. For example, somewhat similar increases in theta power observed in the resting MEG signal recorded from patients with Parkinson's disease, obsessive compulsive disorder and other syndromes have been proposed to arise in the setting of less severe deafferentation and disfacilitation of thalamic neurons producing a 'thalamocortical dysrhymia' syndrome (*Jeanmonod et al., 1996*; *Llinás et al., 1999*). These findings are correlated with theta frequency bursting identified in the thalamus (driven by de-inactivation of low-threshold T-type calcium channels (*Jeanmonod et al., 1996*)) of patients with Parkinson's disease or central pain (*Jeanmonod et al., 1996*; *Llinás et al., 1999*) who lack broad global cerebral deafferentation that characterizes our subject population with disorders of consciousness.

The distinction between global and focal injuries leading to rhythmic theta activity is important because these scenarios predict opposite effects on locally measured EEG activity in the 15–30 Hz range. Specifically, experimental and clinical studies of thalamocortical dysrrhythmia link abnormal increases in theta power to coincident abnormal increases 15–30 Hz activity produced by lateral inhibition in the cortex ('edge effect', Llinás et al., 1999; Llinás et al., 2005). Of note, the linkage of elevated 4–10 Hz and 15–30 Hz activity has been observed in a study of a zolpidem responsive stroke patient subject (*Hall et al., 2010*). *Hall et al.(2010)* studied MEG signals in a fully conscious, alert, and cognitively normal patient subject with mild aphasia and diminished sensorimotor integration ; these investigators reported that zolpidem administration produced similar changes of power suppression in co-localized areas of abnormally increased 4–10 Hz activity and increased 15–30 Hz activity in the resting MEG signal measured near the boundaries of a large stroke lesion within the damaged hemisphere. This suppression of increased high frequency power with zolpidem is exactly opposite to the changes observed in ~15–40 Hz power in our subjects and indicates that a different mechanism underlies the changes seen in the present study, compared to that underlying the resting MEG activity in the Hall et al. subject (*Hall et al., 2010*). In the latter subject, global

cerebral structures, including both thalami, remained intact and the patient's state of consciousness was normal (and there also was no history of a preceding disorder of consciousness). Moreover, in the Hall et al. study, no increases were observed in ~15–40 Hz power outside of MEG channels with abnormal 3–7 Hz power (where ~15–40 Hz suppressed with zolpidem administration). Since the elevated theta and beta activity seen in the Hall et al. study was focal, it is more likely to be due to bursting of a localized set of deafferented thalamic neurons (i.e., thalamocortical dysrhythmia), than to widespread cortical de-afferentation.

To produce thalamocortical dysrhythmia, low-threshold T-type calcium channels must be de-inactivated (*Jeanmonod et al., 1996*; *Llinás et al., 1999, 2005*). This is far more probable to arise in this setting with a more locally deafferented thalamus in largely structurally intact brain. Taken together with the model proposed above, the distinct pattern of spectral changes observed in the Hall et al. patient leads to a prediction concerning the substrate of the pathological ~3–7 Hz EEG oscillations in patients with severe brain injuries. In our subjects—with a widely coherent theta rhythm—we expect that thalamic activity is severely suppressed and not generating widespread bursting activity. In contrast, when focal ~3–7 Hz and ~15–40 Hz oscillations coexist (as in the Hall et al. patient [*Hall et al., 2010*]), we hypothesize that the thalamus is not globally suppressed, and that instead, there is bursting due to focal thalamic deafferentation. The former expectation is supported by metabolic imaging studies in Subject 2, (*Figure 8*, *Table 3*), and other studies in awake severely brain-injured subjects that included direct measurement of thalamic activity that demonstrated loss of central thalamus firing activity (*Giacino et al., 2012*).

In addition to the thalamocortical dysrhythmia model of 6–10 Hz activity, other theoretical models of the expression of these rhythms in the corticothalamic system exist based on network dynamics (e.g., *Ching et al., 2011*; *Drover et al., 2011*; *Hughes et al., 2004*; *Robinson et al., 2002*). However, these models do not include gross alterations in physiological profiles of neurons in severely-injured brains with average neuronal membrane potentials remaining well below firing thresholds and likely do not capture the relevant physiological conditions present in our subjects. While an intrinsic cortical mechanism appears to be sufficient to account for our observations, such independent or interacting contributions of network mechanisms linked to thalamocortical projections ultimately cannot be conclusively excluded.

## Proposed 'mesocircuit' mechanism underlying behavioral facilitation and changes in EEG resting dynamics observed in the ON zolpidem state

We next consider possibilities for how zolpidem might reverse the marked down-regulation of anterior forebrain activity. Two studies have demonstrated increases in cerebral perfusion and metabolism in zolpidem responsive subjects with disorders of consciousness (*Brefel-Courbon et al., 2007*; *Nyakale et al., 2010*). *Brefel-Courbon et al.(2007)* reported broad activation of FDG-PET signal with zolpidem in a patient remaining chronically in minimally conscious state who recovered speech, swallowing and ambulation reliably with zolpidem administration. These changes were marked and bi-hemispheric, demonstrating broad metabolic increases across the frontal lobes, striatum and thalami of both hemispheres (Subject 2 studied here a ~50% reduction in global metabolic rates and relatively reduced frontal, striatal and thalamic metabolic expression consistent with this earlier report as shown in *Figure 8*). Local increases in blood flow within these same anterior forebrain regions have also been reported in the dominant hemisphere of a stroke patient who recovered fluent speech with zolpidem administration (*Cohen et al., 2004*).

A 'mesocircuit' model has been proposed to account for the mechanisms of action of zolpidem in the context of marked down-regulation of the anterior forebrain in the severely injured brain and restoration of activity across the anterior forebrain during recovery (*Brown et al., 2010*; *Schiff, 2010*).

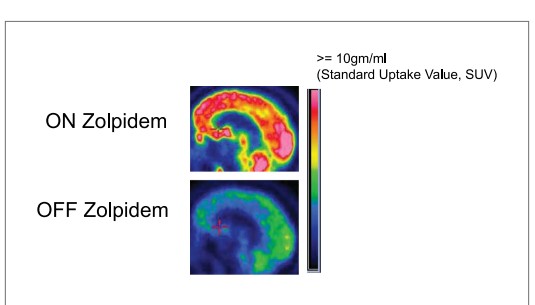

**Figure 8**. FDG-PET measured cerebral metabolism OFF and ON zolpidem for Subject 2. A marked increase in global cerebral metabolic rate is seen with zolpidem administration, average increase is 1.97 times OFF baseline across cerebral structures (see 'Methods' and *Table 3* for regional differences in metabolic rates for selected areas).

**Table 3.** Percent change in regional standardized uptake values with administration of zolpidem

| Region | Left(%) | Right(%) |
|---|---|---|
| Mesial frontal | 139 | 146 |
| Lateral frontal | 166 | 183 |
| Calcarine | 145 | 153 |
| Striatum | 156 | 156 |
| Thalamus | 148 | 127 |

Briefly, this mesocircuit model incorporates the basic observation that all severe brain injuries produce diffuse disfacilitation across corticothalamic systems arising from widespread disconnection or neuronal death (*Adams et al., 2000*; *Maxwell et al., 2006*). As a consequence of their broad and widespread point to point connections, neurons in the central thalamus are progressively more deafferented with severity of structural brain injuries (*Maxwell et al., 2006*). Reduction of corticostriatal, thalamocortical and thalamostriatal outflow following multi-focal deafferentation and loss of neurons across the corticothalamic system likely also results in conditions of sufficient loss of afferent input to the medium spiny neurons (MSNs) of striatum to prevent these neurons from reaching firing threshold because of their requirement for high levels of synaptic background activity (*Grillner et al., 2005*). Loss of active inhibition from the striatum in turn is expected to result in release of tonic activity from neurons of the globus pallidus interna (GPi) that will provide an active inhibition to their synaptic targets including relay neurons of the already strongly disfacilitated central thalamus and possibly also the projection neurons of the pedunculopontine nucleus (*Rye et al., 1996*). Collectively, these effects are expected to produce very broad reductions in global cerebral synaptic activity as reflected in very low cerebral metabolic rates typically measured in severe brain injuries producing disorders of consciousness (reviewed in *Laureys and Schiff, 2012*). The main prediction of this model is that across all etiologies of structural injury, metabolic and functional downregulation of the anterior forebrain should be prominent. The dominance of the ~7.4 Hz activity across fronto-central EEG channels observed here provide support for this prediction.

A primary proposed activating effect of zolpidem is suppression of increased firing of the GPi via a direct effect of zolpidem on GABA-A alpha 1 subtype receptors which are present on all neuronal cell types in the human globus pallidus interna (*Waldvogel et al., 1999*). Zolpidem can also be linked directly to the selective binding of GABA-A alpha 1 receptor subtypes in the neocortex, where it may increase thalamocortical and thalamostriatal outflow indirectly as a result of activation of cortical inhibitory interneuronal networks (*McCarthy et al., 2009*). In addition, zolpidem may activate the striatum where GABA-A currents facilitate alpha and beta (~8–30 Hz) rhythms within the striatum and normal MSN function (*McCarthy et al., 2011*). Finally, a direct stimulatory action of GABAergic agonist at the pyramidal cell axon could produce excitation and persistent gamma-frequency oscillations (*Traub et al., 2003*). Collectively, all of these potential mechanisms of zolpidem's action would result in the marked increase of thalamocortical and thalamostriatal outflow, restoration of corticothalamic and corticostriatal outflow, as well as suppression of tonic inhibition of the central thalamus and possibly the PPN by the GPi.

Zolpidem induced behavioral facilitation and EEG activation in patients has been proposed to directly link to a phenomena generally observed in anesthesia known as 'paradoxical excitation' which arises with propofol and other anesthetics (*Brown et al., 2010*). In the context of anesthesia, broad reductions of background synaptic activity occur during initial sedation. During paradoxical excitation normal subjects demonstrate a brief period of agitation and increased power in the 15–30 Hz frequency range after the initial quieting with light sedation. Paradoxical excitation is more common with GABAergic agonists (*Brown et al., 2010*) and the phenomenon may reflect a similar process of release of thalamocortical outflow. A consistent but less prominent shift of fronto-central activity toward increased beta range frequencies (~15–30 Hz) is initiated by zolpidem in the EEG of awake normal subjects receiving the drug but it is not sustained for more than 30 min before returning to baseline (*Patat et al., 2004*). Thus, it can be expected that all patients administered zolpidem will show similar initial dynamical changes in EEG the early period after drug administration (~30 min). However, in those brain-injured patients who show behavioral facilitation with zolpidem this initial shift of 'paradoxical excitation' is sustained and evolves over time as shown in *Figure 5* and is associated with further dynamical changes (narrowing of increased activity in the 20–35 Hz range). The observations of propofol and zolpidem induced increases in 15–30 Hz activity shows that an initial shift of power in this range is expected in all subjects, it remains unclear how this initial change interacts with the overall increase in cortical activation (as indexed by the loss of the ~7.4 Hz activity) and remains self-sustaining.

Effective behavioral facilitation likely involves a significant recruitment of cerebral activity across many other cortico-cortical and cortical-subcortical pathways that may become self-sustaining and further activating. This interpretation anticipates other contributions from brain arousal systems and ongoing activity within the cerebrum in maintaining the power distribution in the ~20–30 Hz range across the frontal-temporal and central-parietal regions as observed for both Subjects studied over hours (refer to *Figure 5*). Such engagement of the plurality of arousal systems in ongoing wakeful behaviors is a powerful arousal stimulus as demonstrated experimentally even in animals with primary deficits in the orexin wake promoting systems of the hypothalamus (*España et al., 2008*). The evident carry over effects of continued zolpidem use in Subject 2 noted above suggests that restoration of behavior allows additional benefits to be harnessed in drug responders. However, the washout periods here were not varied experimentally, so a more rigorous assessment of this hypothesis awaits future work. The appearance of several posterior EEG channels with increased average center frequency in the 9–10 Hz in both subjects 1 and 2 in association with shorter washout intervals (*Tables 1 and 2*) is also consistent with a time-evolving recruitment of activity; expression of more normal patterns of organized rhythms may require the sustained cortical activation over time, with shorter dosing intervals allowing the effects of each dose to build up background brain activity.

## Methods

Subjects 1 and 2 were admitted to the inpatient neurology unit at New York Presbyterian Hospital—Weill Cornell Campus for several days under an IRB-approved natural history study of recovery following severe brain injuries. For each subject, 24 hr video EEG were obtained while they received a continuation of their at home medication regimens. Zolpidem 10 mg was administered crushed, in solution via syringe one to three times daily by mouth for subject 1 and via percutaneous endoscopic gastrostomy (PEG) tube for subject 2. The resultant EEG recordings were reviewed for each subject, and EEG segments of 3 s duration that were artifact-free in the analyzed channels were identified extracted from the hour before and each of the first 4 hr after each administration of the drug, for each administration studied (four doses in Subject 1, five doses in Subject 2). Subject 3 underwent quantitative behavioral evaluations and study with high density EEG in their home (performed by investigators from University of Liege, Belgium) before and 20 min after taking zolpidem. Zolpidem 10 mg was administered per nasogastric tube. PET studies for ON and OFF drug states were obtained for Subject 1 and 2 but only reported for Subject 2 in 'Clinical histories' here as Subject 1's OFF drug study was severely motion corrupted and reliable measurements could not be obtained. For subject 2, FDG-PET studies in both the zolpidem-naive state and 30–90 min after administration of the drug were obtained 24 hr apart at the same time of day with equivalent fasting serum glucose levels of 81 and 80 mg/dl (see below for details). All procedures done in this study were in compliance with the Declaration of Helsinki.

### Video EEG data acquisition

Subjects 1 and 2 were recorded continuously over 24 hr periods (with occasional interruption for imaging tests) using video EEG (vEEG) at a sampling rate 1024 Hz for subject 1 and 200 Hz for subject 2 using an XLTEK data acquisition system (Xltek-Tech, Ltd., Oakville, Ontario, Canada LH65S1). Silver chloride scalp electrodes (19 electrodes for subject 2, 35 electrodes for subject 1) were affixed with collodion according to a modified international 10–20 system of electrode placement, with the lead at the FPz location as the ground reference. Impedance checks were carried out periodically to ensure impedance less than 5 kiloOhms in all leads.

Subject 3's EEG recordings were acquired using a 256 active electrode high-density EEG system (Electrical Geodesics) sampled at 500 Hz and referenced for acquisition purposes to Cz (the EEG acquisition system default setting). Videotaped recordings were acquired simultaneously and synchronously to the EEG data to confirm the patients' behavior.

### Data analysis

For Subjects 1 and 2 video EEG recordings were manually reviewed on an XLTEK NeuroWorks EEG reviewing station and epochs of 3 s duration without movement artifact were identified that fell within 60 min prior to each recorded zolpidem dose or between 20 and 60 min after each dose. Channels with perceptible muscle artifact were excluded from primary analysis. Additional epochs of 3 s duration with no movement artifact and minimal muscle artifact were identified during the 4 hr subsequent to each zolpidem dose.

All signals were detrended and line noise removed using standard MATLAB functions. We then estimated power spectra using the multi-taper method (*Thomson, 1982*; *Thomson and Chave, 1991*; *Mitra and Pesaren, 1999*; *Mitra and Bokil, 2007*). With five tapers, the effective frequency resolution obtained was 1.7 Hz for 3 s epochs and 2.5 Hz for 2 s epochs. The resulting power spectra were then averaged for all epochs in a condition, and 95% confidence intervals were computed via taper-based jackknife techniques (*Thomson and Chave, 1991*; *Mitra and Bokil, 2007*). All spectral estimates were calculated using Chronux functions, written in MATLAB (*Goldfine et al., 2011*).

Peaks in power were identified by the presence of a maximum in the spectrum within the frequency range of interest (e.g. 6–12 Hz for low frequency peaks).

Time-varying power spectra (spectrograms, *Figure 5*) were derived as follows: for each minute (60 s) of data recorded between one hour prior to and 3–4 hr after zolpidem administration, an epoch of 3 s duration with the least artifact was selected. Power spectra were then estimated as described above for each epoch, and averaging across epochs was not performed.

Coherences were estimated as cross-covariance of the individual signals, normalized by the geometric mean of their power spectra. Nine tapers were used, providing an effective frequency resolution of 3 Hz.

## Significance testing

To determine significant changes between EEG samples obtained from ON and OFF drug conditions we used the Two Group Test (TGT, 12), as implemented by the Chronux toolbox routine, two_group_test_ spectrum (http://www.chronux.org), with a cutoff of p=0.05 by the jackknife method. Because spectral estimates within 2 Hz of each other are correlated by the taper functions, a difference identified by the TGT was only considered significant if it was present for all frequencies contiguously over a range greater than 2 Hz. TGT was applied for all comparisons of spectral power and coherence (*Goldfine et al., 2011*) shown here (see *Tables 1 and 2* for power significance testing of suppression of low frequency peaks for all channels and doses in Subjects 1 and 2, *Figure 2—figure supplement 1*; *Figure 6—figure supplement 1* for example of coherence significance testing across all channel pairs).

### Clinical histories

### Subject 1

A 45 year old man suffered traumatic brain injury due to a fall from a second floor ladder to a concrete surface 5 years prior to the study. At the time of the study, this subject varied in level of behavioral function from a total score on the Coma Recovery Scale-Revised (CRS-R) of 10–14 at baseline to a reliable score of 23 with zolpidem administration (the highest level of function measured by this instrument, a level consistent with emergence from MCS). At baseline (off zolpidem) the subject could stand and transfer from bed to wheelchair with assistance but made no attempts at verbal communication with or without prompting. He followed simple commands only intermittently and could spontaneously demonstrate the functional use of a napkin but no other commonly used objects (cup, toothbrush, comb). On zolpidem the patient regained fluent spoken language, could accurately give his full name, read and write simple words and phrases as well as demonstrate the functional use of common objects as well demonstrating a variety of higher level executive motor skills (e.g., accurate construction of arbitrary block configuration with up to five different block design objects). On zolpidem the subject responded appropriately to wide range of simple yes/no questions (e.g., 'am I pointing to my nose?, do you live in Hawaii?') and general knowledge questions. Additional testing with Mississippi Aphasia Screening Test during two ON zolpidem periods demonstrated a maximum score of 57/100 items on this examination.

### Subject 2

A 32 year old man remained in a minimally conscious state (MCS) for 2 years after a severe mixed traumatic and hypoxic-ischemic brain injury due to near drowning in a motor vehicle accident 9 years prior to the study. At 2 years after injury the subject had a first exposure to zolpidem and demonstrated recovery of spoken language and accurate communication with first dose. Following this observation the patient began a regime of three zolpidem doses daily around the time of meals as use of the medication allowed for sufficient control of deglutition and management of saliva to safely engage in oral feeding. This regime had continued over the 5 year period prior to these studies. Off zolpidem, the subject exhibited prolonged latency of responses to questions or commands, severe dysarthria and dysphagia. He had left-sided hemiparesis and spasticity with flexor posturing and a 1–1.5 Hz

resting tremor of the right upper extremity. After receiving zolpidem the subject was alert, attentive and initiated interaction frequently, with improved verbal fluency and articulation, and with shortened latency of response. He organized complex goal-directed movements of both upper extremities (e.g., formulating an accurate 'knuckleball' baseball pitch using the right hand and arm, throwing the ball across the room). He demonstrated reduced spasticity in the left upper extremity, marked reduction of tremor in the right upper extremity and was able to manipulate small objects such as a pen, spoon and a hairbrush with the right hand. Both loss of oral feeding capacity and spasticity of the right extremity returned after each zolpidem dose had worn off.

### Subject 3

A 55 year old woman suffered a spontaneous hemorrhagic cerebral vascular accident. Prior to zolpidem exposure the patient demonstrated marked arousal impairment and required intensive stimulation to maintain an eyes open awake state. The patient could with vigorous stimulation speak spontaneously emitting only halting expression (stammering, missing words, and frequent phonological and semantic errors). She was confused at baseline and exhibited spatio-temporal disorientation . She also presented visual hallucinations (e.g., 'there is a boy running in the room') and repetitive movements mimicking activities of daily living but out of context (e.g., sewing in the air). The patient was able to demonstrate reproducible responses to verbal command (e.g., squeeze hand) and sustain visual fixation to people in the room but did not demonstrate visual pursuit. Automatic motor behavior such as scratching of nose and passive manipulation of objects occurred spontaneously (e.g., rolling of a ball in the hand). She localized nociceptive stimuli. A yes/no communication system using gesture could be established but without accurate yes/no communication. Following zolpidem administration the patient shifted to an eyes open alert state and showed agitation. Her spontaneous speech rate increased and she formed understandable words in short phrases and initiated interpersonal interactions.

### PET studies for Subject 2

To characterize resting brain metabolism on and off zolpidem, we performed 18-FDG-PET studies on days 2 and 3 for subject 1. This subject had received his last zolpidem dose more than 15 hr prior to the first study and approximately 30–60 min prior to the second study. Both studies were performed on a General Electric Discovery LS PET/CT system (General Electric Company, Fairfield, CT) using a standard resting condition in 2-dimensional (2D) mode. Thirty-five slices were acquired in dynamic high-sensitivity emission mode (matrix size = 128 × 128 × 35, axial field of view = 25 cm, 4.25-mm slice thickness). Attenuation correction and anatomical localization were achieved by performing a helical CT scan in the same session, the images from which were subsequently reconciled to the PET image.

Regions of interest were identified for mesial frontal cortex, lateral frontal cortex, head of the caudate, putamen, thalamus and calcarine cortex. We then calculated standardized uptake values (SUVs) for these regions, as well as globally for each condition. Regional SUVs were then normalized by global averages. Global metabolism increased by a factor of 1.97 on zolpidem in comparison to the off-drug state (*Figure 8*). There was an overall increase in anterior forebrain metabolism, with the largest percent increase occurring in the lateral frontal cortex.

## Acknowledgements

We acknowledge the support of the James S McDonnell Foundation, NIH/NICHD, Belgian National Science Foundation and the Jerold B Katz Foundation for support of these studies. We acknowledge the support of the Weill Cornell and The Rockefeller University's Centers for Clinical and Translational Science Activities for support of this study.

## Additional information

### Funding

| Funder | Grant reference number | Author |
| --- | --- | --- |
| National Institutes of Health | 5R01HD51912 | Shawniqua T Williams, Mary M Conte, Andrew M Goldfine, Bradley Beattie, Jennifer Hersh, Jonathan D Victor, Nicholas D Schiff |

| Funder | Grant reference number | Author |
|---|---|---|
| James S McDonnell Foundation | Consortium for Recovery of Consciousness | Jonathan D Victor, Steven Laureys, Nicholas D Schiff |
| Belgian National Science Foundation | | Quentin Noirhomme, Olivia Gosseries, Steven Laureys |
| Jerold B Katz Foundation | | Nicholas D Schiff |

The funders had no role in study design, data collection and interpretation, or the decision to submit the work for publication.

## Author contributions

STW, NDS, Conception and design, Acquisition of data, Analysis and interpretation of data, Drafting or revising the article; MMC, AMG, QN, OG, SL, Acquisition of data, Analysis and interpretation of data, Drafting or revising the article; MT, Acquisition of data, Analysis and interpretation of data; BB, JDV, Analysis and interpretation of data, Drafting or revising the article; JH, Acquisition of data, Contributed unpublished essential data or reagents; DIK, Drafting or revising the article, Contributed unpublished essential data or reagents

## Ethics

Human subjects: All studies were performed under informed consent provided by subjects' legally authorized surrogates. Subjects 1 and 2 were studied under an Weill Cornell Medical College IRB-approved natural history study of recovery following severe brain injuries. Subject 3 was studied under a separate IRB approved study by the IRB of the University of Liège, Belgium. All procedures were in compliance with the Declaration of Helsinki.

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
