## [Decision Letter]

Thank you for sending your work entitled “Common resting brain dynamics indicate a possible mechanism underlying zolpidem response in severe brain injury” for consideration at *eLife*. Your article has been favorably evaluated by a Senior editor and 3 reviewers.

The following individuals responsible for the peer review of your submission have agreed to reveal their identity: Marcello Massimini (peer reviewer).

The Senior editor and the reviewers discussed their comments before we reached this decision, and the Senior editor has assembled the following comments to help you prepare a revised submission.

*eLife* does not usually provide the complete verbatim reviews to the authors, should they send inconsistent messages. All three reviewers were supportive of the manuscript and found it potentially quite important. But two of the reviewers suggested summarizing the spectral properties of the larger sample of DOC patients studied by the authors to bolster the claim that high power in the 6–10 Hz range is uncommon and may be a predictor of zolpidem response. Adding these data would significantly strengthen the paper.

I am pasting verbatim here a piece of the most critical review for you to consider when preparing your revision, in case these are issues you feel you might address.

“Although this study is of considerable interest, it suffers from some methodologic limitations that should be acknowledged, as well as some limitations in the clarity of exposition.

One methodologic issue is the “performance confound”, frequently discussed in imaging research, but also relevant here. This refers to the fact that when behavioral performance is substantially different in two experimental conditions (in this case on and off zolpidem), and some measures of brain physiology are also different between those 2 conditions, this does not in a straightforward way provide causal evidence that the change in physiologic activity is causally responsible for the change in performance. Since effortful mental activity is typically associated with a shift toward more EEG power at higher frequencies, finding a spectral shift of this kind, accompanying behavioral improvement, is not particularly surprising or illuminating. Indeed, it would be quite surprising if the subjects showed significant cognitive improvement without any change in EEG dynamics. The more relevant question is whether the nature of the EEG power spectrum at baseline is predictive of behavioral improvement with the drug. The authors suggest that it is, by noting that high power in the 6–10 Hz range was not typical of their previously studied subjects. This is a relatively vague statement, however, since we don't know what proportion of an unselected sample of patients with disorders of consciousness display a particular power level in this frequency range, nor whether those subjects were or were not responsive to zolpidem (though, in an unselected series, one would assume that few if any were zolpidem responsive). The most compelling evidence, then, would be a demonstration that significantly fewer patients who fail to respond to zolpidem have a spectral peak at 6–10 Hz and/or that zolpidem does not abolish that peak in non-responders, neither of which is formally reported here.

Another, more modest methodologic issue is the fact that the washout period between doses was apparently not varied experimentally, but rather analyzed after the fact. Thus, the conclusions about carry over effects based on time between doses are not as rigorous as they might otherwise be. In addition to the methodologic issues, the Discussion section is very dense and would benefit from some streamlining. It would be helpful to characterize the similarities and differences between their meso-circuit hypothesis and thalamocortical dysregulation more generally, perhaps in a tabular form or at least a form that makes it easier to pull out the key points, rather than the current rather lengthy prose descriptions.

Finally, the authors may wish to speculate on a phenomenon that is opposite to one they discuss: the development of tolerance to zolpidem's therapeutic effects. In this publication they study 3 individuals who successfully use the medication regularly, and they discuss possible carry over effects. But there also exist patients who respond dramatically to an initial dose of zolpidem but who experience diminished benefit with repeated dosing and restoration of benefit from a prolonged “drug holiday”. Do they have any thoughts about how to account for these 2 pattern of evolution of the drug response?”

---

## [Author Response]

1) *…two of the reviewers suggested summarizing the spectral properties of the larger sample of DOC patients studied by the authors to bolster the claim that high power in the 6–10 Hz range is uncommon and may be a predictor of zolpidem response. Adding these data would significantly strengthen the paper*…

*“The more relevant question is whether the nature of the EEG power spectrum at baseline is predictive of behavioral improvement with the drug. The authors suggest that it is, by noting that high power in the 6–10 Hz range was not typical of their previously studied subjects. This is a relatively vague statement, however, since we don't know what proportion of an unselected sample of patients with disorders of consciousness display a particular power level in this frequency range, nor whether those subjects were or were not responsive to zolpidem (though, in an unselected series, one would assume that few if any were zolpidem responsive). The most compelling evidence, then, would be a demonstration that significantly fewer patients who fail to respond to zolpidem have a spectral peak at 6–10 Hz and/or that zolpidem does not abolish that peak in non-responders, neither of which is formally reported here.*”

This suggestion focuses on our comment that contrasts the common observation of elevated ∼6-10 Hz in patients with severe brain injury, with the less-common finding that this activity is globally coherent. However, while we claim that the global coherence is an essential ingredient of the zolpidem response – as it is a marker of recruitable reserve – we do not intend to claim that its presence will predict a response to zolpidem. As we mention in the Discussion, the rarity of the zolpidem response may relate to other factors – such as genetic variability in the level of GABA alpha1 receptors – and, conversely, that neuromodulators other than zolpidem may be able to harness the same recuritable reserve.

We do, however, clearly see how the original writing and framing of the Discussion would give impression that we might propose that these finding showed a prediction of zolpidem response. So we have now revised the Discussion in several places, and include a “roadmap” that, we hope, better frames the Discussion. As we also now note, determination of the incidence of the globally coherent activity, and whether it is in fact a predictive marker of the zolpidem response, would go far beyond the scope of this work. Key sections of the new text are indicated below:

“...Below we propose a physiological basis for these common resting state dynamics identified by our EEG findings in the OFF drug state that show specific correspondence with known physiological processes…”

2) *“Although this study is of considerable interest, it suffers from some methodologic limitations that should be acknowledged, as well as some limitations in the clarity of exposition. One methodologic issue is the “performance confound”, frequently discussed in imaging research, but also relevant here. This refers to the fact that when behavioral performance is substantially different in two experimental conditions (in this case on and off zolpidem), and some measures of brain physiology are also different between those 2 conditions, this does not in a straightforward way provide causal evidence that the change in physiologic activity is causally responsible for the change in performance. Since effortful mental activity is typically associated with a shift toward more EEG power at higher frequencies, finding a spectral shift of this kind, accompanying behavioral improvement, is not particularly surprising or illuminating. Indeed, it would be quite surprising if the subjects showed significant cognitive improvement without any change in EEG dynamics.”*

We did not think that this would be an issue, since the changes identified in the shape of the power spectrum observed OFF vs ON zolpidem were stable over time, and were identified in analyses that aggregated the spectra over many hours, including periods of both rest and engagement with environment (e.g., Figure 5 shows the spectra of uniformly sampled 3 second segments concatenated over 5 hours for multiple EEG channels in both Subjects 1 and 2.)

However, prompted by the reviewer’s concern, we added a new figure supplement (Figure 2—figure supplement 1) to specifically address this point. As the figure shows, our original findings are unchanged when the spectra are computed only from quiet periods (OFF vs ON zolpidem), or only from periods during environmental stimulation (OFF vs ON zolpidem).

We now state in the manuscript the following:

“Figure 2—figure supplement 1 shows that the changes in spectral shape do not represent a “performance confound”…”

3) “*Another, more modest methodologic issue is the fact that the washout period between doses was apparently not varied experimentally, but rather analyzed after the fact. Thus, the conclusions about carry over effects based on time between doses are not as rigorous as they might otherwise be.”*

We agree with this comment and now note in the Discussion that:

“However, as the washout period here were not varied experimentally a more rigorous assessment of this hypothesis is testable in future work.”

4) “*In addition to the methodologic issues, the Discussion section is very dense and would benefit from some streamlining. It would be helpful to characterize the similarities and differences between their meso-circuit hypothesis and thalamocortical dysregulation more generally, perhaps in a tabular form or at least a form that makes it easier to pull out the key points, rather than the current rather lengthy prose descriptions.”*

We have rewritten many sections of the Discussion, including those that relate to thalamocortical dysrhythmia and the mesocircuit hypothesis. But we do think that the discussion of thalamocortical dysrhythmia is important to show that despite the superficial similarities of the latter entity and the findings reported here, there are some very important differences.

5) “*Finally, the authors may wish to speculate on a phenomenon that is opposite to one they discuss: the development of tolerance to zolpidem's therapeutic effects. In this publication they study 3 individuals who successfully use the medication regularly, and they discuss possible carry over effects. But there also exist patients who respond dramatically to an initial dose of zolpidem but who experience diminished benefit with repeated dosing and restoration of benefit from a prolonged “drug holiday”. Do they have any thoughts about how to account for these 2 pattern of evolution of the drug response?”*

We are aware of this alternative pattern of paradoxical zolpidem (and other sedative agents such as lorazepam) response in some severely brain-injured patients but do not have data that provide helpful guidance into the differences in response profile. One possibility is that such loss of effect occurs at the level of mechanism of activation and could reflect differences in receptor pharmacology, ephaptic transmission phenomena, etc. all specific to the positive allosteric modulation of the GABA alpha 1 subtype receptor. But this is just speculation, and the data in the paper does not bear on it, so we prefer not to bring it up.